# Inflammation during Percutaneous Coronary Intervention—Prognostic Value, Mechanisms and Therapeutic Targets

**DOI:** 10.3390/cells10061391

**Published:** 2021-06-04

**Authors:** Bradley Tucker, Kaivan Vaidya, Blake J. Cochran, Sanjay Patel

**Affiliations:** 1Heart Research Institute, 7 Eliza St., Newtown 2042, Australia; b.tucker@student.unsw.edu.au; 2Sydney Medical School, University of Sydney, Camperdown 2050, Australia; kaivan.vaidya@gmail.com; 3School of Medical Sciences, University of New South Wales, Kensington 2052, Australia; b.cochran@unsw.edu.au; 4Royal Prince Alfred Hospital, Camperdown 2050, Australia

**Keywords:** atherosclerosis, inflammation, cardiovascular disease, percutaneous coronary intervention, angioplasty, myocardial infarction, periprocedural myocardial infarction

## Abstract

Periprocedural myocardial injury and myocardial infarction (MI) are not infrequent complications of percutaneous coronary intervention (PCI) and are associated with greater short- and long-term mortality. There is an abundance of preclinical and observational data demonstrating that high levels of pre-, intra- and post-procedural inflammation are associated with a higher incidence of periprocedural myonecrosis as well as future ischaemic events, heart failure hospitalisations and cardiac-related mortality. Beyond inflammation associated with the underlying coronary pathology, PCI itself elicits an acute inflammatory response. PCI-induced inflammation is driven by a combination of direct endothelial damage, liberation of intra-plaque proinflammatory debris and reperfusion injury. Therefore, anti-inflammatory medications, such as colchicine, may provide a novel means of improving PCI outcomes in both the short- and long-term. This review summarises periprocedural MI epidemiology and pathophysiology, evaluates the prognostic value of pre-, intra- and post-procedural inflammation, dissects the mechanisms involved in the acute inflammatory response to PCI and discusses the potential for periprocedural anti-inflammatory treatment.

## 1. Introduction

Coronary artery disease (CAD) is the leading cause of death globally and affects 6.3% of the living population, with that number being substantially higher in developed countries [1]. The role of inflammation in CAD has gained much attention in recent years, most notably with the Canakinumab Anti-Inflammatory Thrombosis Outcome Study (CANTOS), which demonstrated that targeting inflammation, independent of lipid levels, is an effective strategy to reduce recurrent cardiovascular events [2]. More recently, the Low Dose Colchicine 2 (LoDoCo2) trial and Colchicine Cardiovascular Outcomes Trial (COLCOT) have demonstrated the efficacy of colchicine, a potent and readily available anti-inflammatory medication, for the secondary prevention of CAD [3,4]. Despite dramatic improvements in the medical management of CAD and its risk factors, percutaneous coronary intervention (PCI) remains a primary tool for the treatment of CAD. With the use of PCI on the rise, particularly for elective procedures [5], it is essential we fully understand the risks associated with this procedure and the underlying mechanisms. Periprocedural complications, such as myocardial injury and myocardial infarction (MI), are not uncommon, although their precise pathophysiological mechanisms are poorly understood. More specifically, the significance of periprocedural inflammation and its impact on both short- and long-term outcomes remains ill-defined. Therefore, in this review we (1) provide an overview of periprocedural myocardial injury and MI epidemiology and pathophysiology, (2) highlight the prognostic value of pre- and post-procedural inflammation, (3) dissect the mechanisms involved in the inflammatory response to PCI and (4) discuss the potential for anti-inflammatory drugs in the periprocedural period.

## 2. Periprocedural Myocardial Injury and Infarction

### 2.1. Definition

The definition of periprocedural myocardial injury and MI is a contentious issue and has been subject to much change over the last two decades [6]. The most recent definition of periprocedural MI comes from the Fourth Universal Definition of Myocardial Infarction [7]. This consensus document defines a periprocedural MI as an elevation of cardiac troponin >5 times the 99th percentile upper reference limit in patients with a normal preprocedural troponin level. In those with a stable (≤20% variation) or falling preprocedural troponin level, the postprocedural value must rise by >20% and still be >5 times the 99th percentile upper reference limit. To meet the diagnostic criteria for a periprocedural MI, this biochemical evidence must be combined with clinical evidence of new myocardial ischaemia (i.e., ischaemic ECG changes, pathological Q waves, imaging evidence of loss of viable myocardium and/or angiographic findings consistent with a procedural flow-limiting complication (slow flow or no-reflow)). The term periprocedural myocardial injury is used to define the cohort of individuals who have a postprocedural troponin elevation but do not meet the diagnostic criteria for a periprocedural MI: elevation of cardiac troponin >99th percentile upper reference limit (for those with a normal pre-procedure value) or a rise in cardiac troponin of >20% of the pre-procedure value (for those with an elevated, yet stable or falling preprocedural value) [7]. Other definitions have been proposed, yet they remain less universally accepted [8,9]. 

### 2.2. Epidemiology and Prognostic Significance

The incidence of periprocedural myocardial injury and MI varies greatly depending upon which definition is used [6,10]. A recent analysis of data from the SYNTAX trial compared the rate and prognostic significance of different definitions of periprocedural MI [10]. In this population, the rate of periprocedural MI varied from 2.7% to 6.0%, yet regardless of definition, periprocedural MI was associated with an increased risk of major adverse cardiovascular events (MACE) at 5 years and all-cause mortality at 10 years. In a comprehensive analysis of patients with stable CAD who underwent elective coronary stenting, Zeitouni et al. (2018) utilised the Universal Definition of Myocardial Infarction and reported the incidence of periprocedural myocardial injury and MI as 22% and 7%, respectively [11]. Similar to the results of the SYNTAX trial, both periprocedural myocardial injury and MI were associated with increased 30-day MACE, although only periprocedural MI predisposed to cardiovascular events at 1 year [11], a finding corroborated by results of the EXCEL trial [12]. 

### 2.3. Pathophysiology

The pathophysiology of periprocedural MI can be divided into two categories, as per Herrmann’s seminal review [13]. Proximal periprocedural MIs occur in close proximity to the target lesion and are most commonly a result of side branch occlusion (SBO). Distal periprocedural MIs occur in the distal perfusion territory of the target coronary artery and are largely the result of microembolisation and microvascular obstruction. Of the two, distal periprocedural MI is the most common [14].

#### 2.3.1. Proximal Periprocedural MI 

SBO is primarily the result of physical arterial manipulation; hence, predictors of SBO relate to physical properties of the lesion, location and stent type [15]. SBO occurs secondary to plaque compression and mechanical straightening of the vessel, which in turn causes luminal obstruction of adjacent side branches either by shifting of atherosclerotic plaque into the origin of the side branch or by altering the angle of the side branch origin [15]. As such, proximal periprocedural MIs are largely the result of physical plaque manipulation leading to vessel occlusion, in which there is a limited role for inflammation. 

#### 2.3.2. Distal Periprocedural MI 

Distal embolisation of plaque material with subsequent microvascular obstruction is the dominant cause of periprocedural MI [14] and, in contrast to proximal periprocedural MI, there is growing evidence to support an essential role of inflammation in its pathogenesis. Mechanical disruption of atherosclerotic plaque promotes embolisation of plaque material, in particular platelet aggregates, cholesterol crystals, leukocytes and cellular debris. Embolised material travels downstream and contributes to microvascular obstruction via a combination of physical arterial occlusion, local platelet activation, vasospasm, oxidative stress and inflammation. This cascade typically manifests angiographically as coronary no-reflow (reduced postprocedural Thrombolysis In Myocardial Infarction (TIMI) flow or increased TIMI frame count) and/or biochemically as an elevation in postprocedural cardiac enzymes (periprocedural myocardial injury or MI). Coronary no-reflow refers to myocardial hypoperfusion in the setting of a patent epicardial coronary artery and is a common cause of periprocedural MI [16,17]. Moreover, the presence of transient or persistent coronary no-reflow is independently associated with a greater risk of short- and long-term mortality [18,19]. 

Advances in intracoronary imaging have facilitated the characterisation of plaque most susceptible to intraprocedural microembolisation. On intravascular ultrasound, large amounts of attenuated plaque have been repeatedly associated with an increased risk of coronary no-reflow [20,21] and is an independent predictor of periprocedural MI [22]. Two prospective studies involving patients with stable angina [23] and ST-segment elevation MI (STEMI) [24] have demonstrated a positive linear relationship of necrotic core volume with maximum postprocedural creatine kinase-MB (CK-MB) elevation. The necrotic core of atherosclerotic lesions contains fragile tissue and pro-inflammatory debris, such as cholesterol crystals, foam cells and microcalcifications [25]. Furthermore, the presence of attenuated plaque has been associated with increased systemic inflammation [26]. When taken together, these imaging studies suggest that local and systemic inflammation may predispose to periprocedural complications by enhancing plaque fragility and thus the risk of microembolisation. 

## 3. Preprocedural Inflammation and Prognosis

Increased preprocedural inflammation is associated with a higher incidence of short- and long-term complications [27]. Table 1 highlights epidemiological evidence that defines the relationship of the preprocedural inflammatory state with PCI outcomes. 

### 3.1. C-Reactive Protein 

C-reactive protein (CRP) is an acute-phase reactant primarily produced by the liver in response to interleukin-6 (IL-6). CRP is an independent predictor of primary and secondary cardiovascular events [59] and reducing CRP has been shown to improve periprocedural outcomes [60]. Several observational studies have demonstrated the predictive value of preprocedural CRP for periprocedural MI. In a cohort of 500 participants who underwent nonemergent PCI, a preprocedural CRP > 3 mg/L was associated with a 2.4-fold increase in the risk of periprocedural MI [28]. Similar results have been reported in a host of other studies [28,29,30,31,32,33,34]. Zhao et al. [34] further added that the association of preprocedural CRP with periprocedural MI was more pronounced in non-smokers and in individuals without renal insufficiency, suggesting that inflammation secondary to a coronary pathology rather than smoking or another chronic disease is primarily responsible for this relationship. In a large population of patients with acute coronary syndrome (ACS) and stable CAD who underwent nonurgent PCI, there was a positive and linear relationship of preprocedural CRP with postprocedural CK-MB [33]. 

Preprocedural CRP has also been shown to be predictive of coronary no-reflow. In a cohort of 1,140 patients with STEMI, of whom 108 developed coronary no-reflow, increased preprocedural CRP was associated with increased odds of no-reflow [18]. This relationship has since been corroborated in several large independent cohorts [35,61]. Furthermore, in patients with chronic total occlusion, an elevated preprocedural high-sensitivity CRP (hsCRP) is an independent predictor of coronary slow-flow and no-reflow [36]. 

Preprocedural CRP levels are also predictive of longer-term complications such as recurrent MI, restenosis and mortality. In a secondary analysis of the PROVE IT–TIMI 22 trial, which included 4162 participants with ACS of whom 70% underwent PCI for their index event, each 1 standard deviation increase in baseline CRP was associated with a 30% increased risk of recurrent MI [62]. In a smaller population of 167 participants with stable CAD who had drug-eluting stents implanted, those with a preprocedural CRP level in the highest tertile were at a 3-fold higher risk of restenosis at 8 months compared to those in the lowest tertile [37]. Similarly, high preprocedural CRP levels have been shown to confer an increased risk of Q-wave MI and death [38,39]. This long-term prognostic significance of preprocedural CRP has been highlighted in a recent systematic review and meta-analysis involving 34,367 participants [27]. In this pooled analysis elevated preprocedural CRP levels were associated with an increased incidence of MACE, all-cause mortality, coronary revascularisation and clinical restenosis. In contrast, preprocedural CRP is a poor predictor of future ventricle function. Over three separate trials in STEMI patients undergoing primary PCI, Świątkiewicz et al. found that CRP upon admission was not predictive of future left ventricular remodelling, left ventricular systolic dysfunction or heart failure hospitalisations [63,64,65].

### 3.2. Haematological Parameters 

Similar to CRP, leukocyte count is a readily available marker of inflammation that is widely used in clinical practice. Leucocytosis is a well-known risk factor for future cardiovascular events in both individuals with and without clinically evident CAD [66]. Leucocytosis can predispose to periprocedural complications by causing endothelial cell injury, hypercoagulability, microvascular plugging and plaque fragility [67]. As such, various preprocedural haematological parameters have been investigated for their prognostic value.

Despite being a non-specific marker of inflammation, preprocedural total leukocyte count is an independent predictor of periprocedural MI. In a cohort of 880 participants with stable CAD who underwent elective PCI, the incidence of periprocedural MI progressively increased from tertile 1 to 3 of preprocedural total leukocyte count, with reported incidences of 6.5, 9.6 and 13.8%, respectively [40]. This link is supported by a novel study from Aronow et al. [41] who reported a positive correlation between preprocedural leukocyte count and intraprocedural microembolisation. Aronow and colleagues included 43 participants undergoing carotid stenting with simultaneous monitoring of microembolic signals in the ipsilateral middle cerebral artery via transcranial Doppler. Even after adjustment for baseline characteristics and concomitant medication use, increasing leukocyte count was a significant predictor of embolisation during stenting. Therefore, it is conceivable that leukocytosis increases plaque fragility thereby predisposing to intraprocedural microembolisation with subsequent microvascular injury/obstruction and myonecrosis. In fact, in a study of 1,217 participants with STEMI each 10^3^ increment in total leukocyte count was associated with 6.6% higher odds of coronary no-reflow [35], substantiating results of an earlier study [42]. In the longer term, an elevated preprocedural total leukocyte count is associated with an increased risk of all-cause mortality and recurrent MI following PCI [43,44]. As total leukocyte count is a crude marker of the inflammatory response, further studies have helped elucidate specifically which immune cells are responsible for this relationship. 

An elevated eosinophil count is detrimental in the long-term. However, a high preprocedural eosinophil count is not associated with periprocedural MI and is in fact a protective factor within the first 6 months post-PCI [45,46]. In contrast, an elevated preprocedural neutrophil count has been linked to adverse periprocedural outcomes. Several observational studies have shown that either neutrophilia or an elevated neutrophil to lymphocyte ratio is associated with an increased risk of coronary no-reflow in patients undergoing primary PCI [47,48,49]. Furthermore, an elevated preprocedural neutrophil to lymphocyte ratio is an independent predictor of three-year mortality and MACE [50]. The accurate predictability of the neutrophil to lymphocyte ratio relates to its reflection of both opposing sides of the immune system—neutrophilia representing active nonspecific inflammation and lymphopenia representing poor overall health status and physiological response to stress [68].

The lymphocyte to monocyte ratio represents another haematological marker of inflammation in the periprocedural period. In two separate studies, a lower lymphocyte to monocyte ratio was associated with higher odds of coronary no-reflow [51,52]. More specifically, Kurtul et al. [52] reported that a preprocedural lymphocyte to monocyte ratio < 2.292 had a 76.3% sensitivity and 72.5% specificity in predicting coronary no-reflow. In a separate study, decreased preprocedural lymphocyte to monocyte ratio increased the risk of both in-hospital and long-term MACE [53]. A higher monocyte to high-density lipoprotein cholesterol ratio has also been identified as a risk factor for coronary no-reflow [51]. Although not routinely or easily measured, increased preprocedural platelet–leukocyte aggregates are a known risk factor for coronary no-reflow. Platelet–leukocyte aggregates are a measure of platelet activity and are essential for not only thrombus formation but also propagation of the inflammatory cascade [69]. By measuring platelet–leukocyte aggregates in the peripheral blood of participants prior to primary PCI, Ren and colleagues [54] demonstrated that elevated levels of circulating platelet–neutrophil and platelet–monocyte aggregates were highly predictive of coronary no-reflow. 

### 3.3. Other Inflammatory Markers 

Myeloperoxidase (MPO) is a predominantly neutrophil-derived enzyme responsible for catalysing the formation of reactive oxygen species, which are essential in the innate immune and acute inflammatory responses [70]. In patients with an acute MI who undergo PCI, the concentration of MPO and interleukin-8 (IL-8; a chemotactic protein specific for neutrophils) are markedly increased in the culprit artery compared to the peripheral circulation [55]. Furthermore, the extent of MPO release in the culprit artery directly correlates with the corrected TIMI frame count [55]. This suggests that IL-8 is released from the culprit lesion during an acute MI resulting in rapid neutrophil recruitment and degranulation, which directly contributes to microvascular obstruction. In a separate study, higher concentrations of MPO in the culprit artery of acute MI patients during PCI was associated with a greater extent of microvascular obstruction at 1 week and 6 months post-PCI [56]. Additionally, rapid inhibition of MPO after an MI in mice has been shown to improve both short- and long-term cardiac function [71], likely by attenuating the deleterious effects of MPO on the microvasculature.

Monocyte chemoattractant protein-1 (MCP-1), as the name suggests, is the primary mediator of monocyte recruitment and activation. MCP-1 is essential in the pathogenesis and progression of atherosclerosis and elevated local coronary levels of MCP-1 have been associated with an unstable phenotype [72,73]. In a population of 192 participants with STEMI, elevated preprocedural MCP-1 levels were shown to be predictive of coronary no-reflow and 3-year mortality [57]. After adjustment for potential confounding factors, each 1 pg/mL increase in serum MCP-1 was associated with 5% higher odds of no-reflow. As MCP-1 levels reflect lesion stability, increased preprocedural MCP-1 levels likely identify those plaque most susceptible to microembolisation. Moreover, increased MCP-1 would augment the inflammatory response to PCI and predispose the microvasculature to injury. 

Lipoprotein-associated phospholipase A2 (Lp-PLA2) is a novel marker of inflammation, with serum levels shown to correlate with the extent of CAD [74]. Soluble Lp-PLA2 mostly binds to low-density lipoprotein (LDL) cholesterol resulting in the generation of oxidised LDL (oxLDL), which induces endothelial dysfunction and leukocyte activation [75]. In a recent study, of 265 participants who underwent elective PCI, preprocedural Lp-PLA2 level was directly proportional to the risk of periprocedural MI [58]. Using the optimal cut-off point of 185 ng/mL, the sensitivity and specificity of preprocedural Lp-PLA2 for predicting periprocedural MI were 65 and 76%, respectively. 

## 4. PCI-Induced Inflammation and Prognosis

Periprocedural increases in inflammation generally reflect PCI-induced vascular injury and are associated with a poor prognosis. In patients with stable CAD who underwent elective PCI, Gach and colleagues [76] reported that a significant increase in hsCRP from pre- to 24 h post-PCI (defined as ΔhsCRP ≥ 3 mg/L) was a robust independent predictor of future MACE in both the short- and long-term. Over a median follow up period of 6.5 years the MACE rate was 37% vs. 3.4% in the ΔhsCRP ≥ 3 mg/L and ΔhsCRP < 3 mg/L groups, respectively. Moreover, the authors reported that the ΔhsCRP had a higher predictive value for future MACE than either the preprocedural or postprocedural hsCRP levels alone [76]. Likewise, Saleh et al. [77] reported that those with the greatest ΔhsCRP had the greatest risk of periprocedural MI and future MACE. In two other studies of patients with stable CAD, a greater change in CRP from pre- to 24 h post-PCI as well as a prolonged CRP elevation following PCI were both associated with a greater risk of restenosis at 6 and 12 months [78,79]. The periprocedural change in CRP has also been shown to positively correlate with the extent of postprocedural neointimal hyperplasia [80]. Finally, a positive relationship has been reported between the change in the neutrophil to lymphocyte ratio from pre- to post-PCI and future MACE [81].

## 5. Postprocedural Inflammation and Prognosis

The extent of postprocedural inflammation reflects a combination of atherosclerosis-associated inflammation as well as PCI-induced vascular injury. The level of inflammatory markers in both the short- and long-term postprocedural period has important prognostic value. In the acute postprocedural period (i.e., measured within 48 h of PCI) hsCRP directly correlates with postprocedural high sensitivity troponin-I [82]. In patients with ACS, a higher postprocedural CRP concentration is associated with enhanced progression of coronary atherosclerotic disease in the subsequent 6–12 months [83]. Another emerging biomarker of cardiovascular inflammation is pentraxin 3. From the same family as CRP, pentraxin 3 is an acute phase inflammatory glycoprotein synthesised primarily by endothelial cells and leukocytes [84]. Recently, an elevated postprocedural pentraxin 3 level was shown to be independently associated with a greater risk of future MACE in a cohort of patients with stable CAD [85].

Although preprocedural inflammation is not a strong predictor of long-term heart failure, postprocedural inflammation is. In STEMI patients treated with primary PCI, the CRP levels at 24 h after admission and at discharge are independent predictors of future cardiac function. Elevated CRP levels during hospitalisation for acute MI are associated with higher rates of left ventricular remodelling and systolic dysfunction at 6-months post-MI [45,47,86]. Similarly, those with an elevated CRP level at 24 h post-admission, discharge and 1 month post-MI were all at a substantially increased risk of future heart failure hospitalisations [64]. Stumpf et al. further demonstrated a robust negative correlation (*r* = −0.56) between maximum post-MI CRP and left ventricular ejection fraction [87]. In a small cohort of 61 STEMI patients who underwent primary PCI and developed heart failure in the following 48 h, a higher concentration of postprocedural IL-8 was associated with less improvement in left ventricular function over the following 6 weeks [86]. The prognostic value of postprocedural inflammation is further highlighted by the results of the CANTOS and SOLID-TIMI 52 trials. In the CANTOS trial, approximately 65% of participants had a history of PCI, most of which was secondary to their index event [88]. Over the median follow-up of 3.7 years, those with hsCRP or IL-6 levels in the highest tertile at baseline were at a two-fold increased risk of heart failure hospitalisation compared to those in the lowest tertile. More importantly, those randomised to canakinumab (a human monoclonal anti-IL-1β antibody) who achieved an on-treatment hsCRP concentration of <2 mg/L reported significantly less heart failure hospitalisations and heart failure-related mortality—an effect which was obvious in both individuals with and without a history of heart failure [88]. In a subset of 4939 participants within the SOLID-TIMI 52 trial, of whom approximately 75% underwent PCI for their index event, IL-6 levels were measured at baseline [89]. Over the median follow-up period of 2.5 years, there were 182 heart failure hospitalisations and, after adjusting for baseline characteristics and prognostic factors, those in the highest IL-6 quartile had a three-fold increased risk of heart failure hospitalisation compared to those in the lowest quartile [89]. Although primary PCI is the mainstay of treatment for acute MI, these data indicate that an augmented inflammatory response within the periprocedural period predisposes to future heart failure. 

Residual inflammatory risk (RIR) describes the inflammatory status of individuals with cardiovascular disease and is most commonly defined as a hsCRP > 2 mg/L. The clinical importance of such risk was highlighted in a secondary analysis of the CANTOS study, whereby canakinumab (administered subcutaneously every 3 months) was shown to be most effective in those who achieved a reduction in hsCRP below 2 mg/L after their first dose [90]. However, it is important to note that in this study the beneficial anti-inflammatory effect of canakinumab was outweighed by the increased risk of infection. Until the recent publication of two large retrospective cohort studies [91,92], the concept of postprocedural RIR had not been addressed. Both studies involved a large cohort of both stable CAD and ACS patients drawn from the PCI registry of Mount Sinai Hospital (NY, USA). Kalkman et al. [92] published the first of these two studies and included 7026 participants who underwent PCI and had hsCRP measured at baseline and approximately 1 year later. Most notably results of this study demonstrated that those patients with persistently high RIR (two hsCRP values > 2 mg/mL) were at the greatest risk of one-year MACE and mortality. Utilising multivariable Cox regression with persistently low RIR (two hsCRP values < 2 mg/mL) as the reference point, persistently high RIR had an adjusted hazard ratio of 3.2 for 1-year all-cause mortality and 1.6 for 1-year MI. This association of persistently high RIR and mortality or MI was present in both ACS and stable patients who underwent PCI. The prevalence of persistently high RIR in this population was 38%. Similarly, increased RIR (one CRP < 2 mg/mL then one > 2 mg/mL) was associated with an increased risk of MI at 1-year, whereas the rate of death at 1 year was lower in the attenuated RIR group (one CRP > 2 mg/mL then one < 2 mg/mL), although the latter difference failed to reach statistical significance. The second study published by Guedeney et al. [91] assessed the prognostic significance of postprocedural RIR among 3013 patients with controlled cholesterol risk (LDL cholesterol ≤ 70 mg/dL). Corroborating results of the previous study, Guedeney et al. demonstrated that persistently high RIR and increased RIR were associated with higher rates of one-year MACE and mortality, whereas there was no such risk for those in the attenuated RIR group. In this ‘cholesterol-controlled’ cohort the prevalence of persistently high RIR was similar to the previous study at 34%. When taken together these studies show that a sustained inflammatory response following PCI is common and independent of cholesterol control and is associated with a greater risk of future cardiovascular events and mortality. In contrast, attenuated RIR was not associated with such endpoints, suggesting that anti-inflammatory interventions in the peri- and post-procedural period may provide a mortality benefit. 

## 6. Mechanistic Link between Inflammation and Periprocedural MI

CAD is a chronic inflammatory condition that is exacerbated during ACS. When compared to individuals with stable CAD, patients with ACS have markedly elevated levels of interleukin-1β (IL-1β), interleukin-18 (IL-18) and IL-6 [93]. Moreover, the concentration of these pro-inflammatory cytokines is higher in patients with stable CAD compared to individuals without evident coronary disease [93]. Despite being a therapeutic intervention, PCI further augments the inflammatory response in these patients. There are two major sources of inflammation in the periprocedural period—the underlying coronary disease and PCI itself. As the aforementioned evidence clearly demonstrates, high levels of preprocedural inflammation are associated with a worse prognosis. Moreover, an augmented inflammatory response to or induced by PCI is disadvantageous. Herein, we review clinical and preclinical data to highlight the possible mechanisms by which PCI elicits this inflammatory response. 

The inflammatory response to PCI is largely driven by direct mechanical trauma induced by balloon inflation and/or stent deployment which can (1) disrupt plaque leading to microembolisation and release of prothrombotic/proinflammatory material; (2) damage the coronary endothelium (Figure 1). Microembolisation can occur during all stages of PCI and the number of microemboli released directly correlates with the extent of postprocedural myonecrosis [94]. Microembolisation contributes to microvascular obstruction and ultimately myocardial injury by occluding distal vessels and triggering a prothrombotic and proinflammatory response. Charron et al. [95] highlighted the impact of microembolisation on inflammation by utilising a coronary no-reflow model in pigs. In this study, biologically inert polystyrene microspheres were injected into the left anterior descending artery via a transit catheter and resulted in rapid formation of neutrophil-platelet aggregates and release of tumour necrosis factor-α (TNF-α). This pro-inflammatory effect of microembolisation was seen even at relatively low microsphere doses that did not affect angiographic coronary flow. These data demonstrate that even a small degree of embolisation can have a marked effect on the inflammatory cascade, which can ultimately amplify the effect of microemboli and contribute to microvascular obstruction. Endothelial injury is another unavoidable consequence of PCI. Markers of endothelial damage/dysfunction such as von Willebrand factor, soluble E-selectin and circulating endothelial cells are increased in peripheral venous blood within as little as 15 min post-PCI [96]. Loss of, or damage to, the coronary endothelial layer not only predisposes to thrombus formation and vasospasm but also potentiates the inflammatory response [97]. In response to injury the endothelium activates protein kinase C and nuclear factor-κB (NF-κB) signalling, which in turn augments the expression of cellular adhesion molecules, chemokines and cytokines [98,99]. This promotes the recruitment, activation and transmigration of leukocytes, particularly neutrophils, to the procedure site [99]. 

In those with an acute MI, myocardial reperfusion injury represents an additional source of inflammation in the periprocedural period. During ischaemic injury the myocardium is deprived of oxygen and essential nutrients ultimately resulting in a host of localised biochemical and metabolic changes, including overt lactic acidosis, mitochondrial dysfunction and cellular swelling [100]. Rapid restoration of blood flow by primary PCI reactivates mitochondrial function, specifically the electron transport chain, leading to the generation of reactive oxygen species (ROS). The particulars of ROS generation in the setting of ischaemia-reperfusion injury have been extensively reviewed elsewhere [101]. Cardiomyocyte death in combination with ROS generation leads to the upregulation of a host of proinflammatory cytokines and chemokines, including complement fragments, TNF-α, IL-1, IL-6 and IL-8. ROS-induced cytokine and chemokine expression is at least in part mediated by NF-κB signalling [102,103]. This proinflammatory milieu stimulates neutrophil recruitment and activation, resulting in a positive feedback loop causing further myocardial injury and production of proinflammatory products [104]. The essential role of inflammation in reperfusion injury has been highlighted in numerous preclinical studies which have demonstrated that attenuating inflammation limits myocardial reperfusion-induced damage. In mice exposed to 30 min of myocardial ischaemia followed by reperfusion, antagonism of Toll-like receptor (TLR) 2 (a pattern recognition receptor located upstream of NF-κB) only 5 min prior to reperfusion was shown to limit infarct size and preserve cardiac function [105]. Similar results have been achieved by inhibiting other components of the inflammatory cascade, such as the nucleotide-binding oligomerisation domain-like receptor, pyrin domain-containing 3 (NLRP3) inflammasome, MCP-1 and TNF-α [106,107,108,109].

Neutrophils are the main immune cell involved in mediating inflammation in the periprocedural period. They contribute to myocardial injury by releasing pre-formed granule proteins, shedding microparticles and forming NETs. As previously discussed, neutrophil-derived MPO increases the production of ROS thereby contributing to lipoprotein modification and endothelial damage [110]. LL37 and α-defensins, two other granule proteins secreted by neutrophils, are primarily responsible for monocyte recruitment and platelet activation [110]. Neutrophil-derived microparticles, small vesicles originating from the plasma membrane, are significantly increased following PCI [111]. These microparticles can activate the complement cascade, endothelial cells and tissue factor [111]. There is growing evidence to support the role of NETs, the terminal product of neutrophil activation, in the pathogenesis of periprocedural MI. In a study of 111 patients with STEMI [112], culprit lesion NET burden during primary PCI was inversely associated with ST-segment resolution and directly associated with infarct size as measured by both CK-MB and cardiac magnetic resonance imaging (MRI). More recently we have shown that PCI, particularly in patients with ACS, results in a rapid increase in the concentration of NETs in the coronary circulation [113]. Mechanistically the rapid rise in NET concentration occurs due to PCI-induced disruption of the plaque–thrombus interface causing the release of pre-activated neutrophils from vulnerable plaque and/or PCI-induced de novo neutrophil activation. This theory is supported by an earlier study that demonstrated substantially greater release of neutrophil-derived microparticles and MPO into the coronary circulation after PCI of unstable vs. stable lesions [111]. NETs consist of a complex network of extracellular DNA, histones, neutrophil elastase and MPO [114]. NETs have a prominent pro-inflammatory effect which is mediated by the formation of neutrophil–platelet aggregates, endothelial injury and release of inflammatory cytokines from macrophages, lymphocytes and endothelial cells. Furthermore, NETs promote thrombosis by providing a scaffold for platelets, red blood cells and other pro-coagulative molecules as well as by activating the intrinsic coagulation cascade [115]. Pre-clinical evidence has demonstrated that NETs accumulate in the myocardium during ischemia/reperfusion injury and that inhibiting NET formation by treatment with DNase-base enzymes improves coronary microvascular patency and limits infarct size [116]. Therefore, inhibiting NET formation or clearing intravascular NETs may reduce periprocedural inflammation and improve outcomes.

Platelets are another source of inflammation in the periprocedural period. Following vascular injury, such as that induced by PCI, platelets adhere to the exposed subendothelial matrix and become activated. Despite universal preprocedural antiplatelet treatment, high residual platelet activity in the postprocedural period remains common and is associated with an increased risk of short- and long-term MACE [117,118,119]. Activated platelets secrete thromboxane A2 and adenosine diphosphate, which facilitate thrombus formation [119]. Beyond their well-known role in haemostasis, platelets play an important role in the inflammatory response by secreting cytokines and chemokines, augmenting adhesion molecule expression and modulating leukocyte function. α-granules are the most abundant storage granule within platelets [120]. Along with other proteins, α-granules contain a plethora of chemokines responsible for promoting neutrophil and monocyte recruitment, including platelet factor 4 (PF4), IL-8, macrophage inflammatory protein-1α and RANTES [120,121]. Of these, PF4 is the most abundant. In both neutrophils and monocytes, PF4 promotes recruitment, adhesion to endothelial cells, cytokine release and ROS production [122,123], the latter of which may contribute to reperfusion injury in the setting of primary PCI. Another key constituent of α-granules is P-selectin. Expression of P-selection on the surface of activated platelets is the major mediator of platelet–leukocyte aggregates [124]. Platelet–leukocyte aggregates not only facilitate leukocyte extravasation by enhancing their adhesive properties but also amplify leukocyte activity. Shortly after PCI, circulating levels of monocyte– and neutrophil–platelet aggregates increase dramatically [124]. Platelet–neutrophil interactions augment the production of ROS and NETs, both of which are associated with adverse outcomes [125]. Similarly, platelet–monocyte interactions promote phenotypic changes in monocytes with a shift toward a more proinflammatory subtype which displays enhanced cytokine production and a propensity for foam cell formation [126,127,128]. 

The inflammatory response to PCI is largely regulated by the NLRP3 inflammasome. To fully function the NLRP3 inflammasome requires two levels of stimulation—priming and activation. Priming of the NLRP3 inflammasome requires signalling via receptors that activate NF-κB-mediated transcription, such as TLRs or IL-1 receptors [129]. Many damage-associated molecular patterns or pathogen-associated molecular patterns can prime the inflammasome. One of the most extensively studied mediators is oxLDL. oxLDL is a well-known inducer of TLR signalling and is acutely increased following PCI [130]—mostly likely being released during plaque manipulation and/or generated secondary to MPO release from activated neutrophils. In ACS patients, the inciting event such as plaque rupture can be sufficient to prime the inflammasome in circulating leukocytes [131,132]. It has been previously shown that, in ACS patients, the inflammasome is primed in peripheral monocytes, releasing high concentrations of IL-1β after ex vivo adenosine triphosphate stimulation [131]. Hence, ACS patients are more susceptible to inflammasome activation and an augmented inflammatory response to PCI. Priming of the NLRP3 inflammasome involves nuclear translocation of the transcription factor NF-κB thereby inducing the expression of pro-IL-1β, pro-IL-18 and NLRP3 [133]. Several stimuli can activate the NLRP3 inflammasome, most notably cholesterol crystals and NETs. Phagocytosis of cholesterol crystals liberated during PCI can induce lysosomal rupture within leukocytes, thus activating the NLRP3 inflammasome [129]. Similarly, NETs can activate the NLRP3 inflammasome, at least in part by activation of the P2X7 receptor [134]. Upon activation, NLRP3 combines with the adaptor molecule apoptosis-associated speck-like protein containing a caspase recruitment domain and pro-caspase-1. Autoproteolysis of pro-caspase-1 produces active caspase-1, which in turn catalyses the processing of pro-IL-1β and pro-IL-18 into their active forms [129]. Signalling via the IL-1 receptor and IL-1 receptor accessory protein, IL-1β activates NF-κB and mitogen-activated protein kinase signalling pathways [133]. Acting via NF-kB, IL-1β functions to stimulate its own production and the production of other secondary inflammatory mediators, such as IL-6 [135]. PCI has been shown to induce IL-1β expression in endothelial, inflammatory and adventitial cells within 1 h [136]. The role of NF-κB signalling in PCI-mediated injury was confirmed by Zeng and colleagues [137] who showed that ischaemic/reperfusion injury in rabbits causes the upregulation of various adhesions molecules, chemokines and cytokines. In the same study, inhibition of NF-κB suppressed the production of these inflammatory proteins, ultimately reducing neutrophil infiltration in the no-reflow region and inhibiting expansion of the no-reflow zone. 

Produced downstream of the inflammasome, IL-6 is the primary inducer of hepatic CRP synthesis and secondary mediator of the inflammatory cascade [138]. IL-6 is unequivocally involved in CAD from atherogenesis through plaque rupture and thrombus formation [139,140]. Several studies have demonstrated a rapid rise in IL-6 levels following PCI. In a small population of 32 participants undergoing elective PCI, Hojo et al. [141] reported a substantial increase in the coronary sinus concentration of IL-6 immediately following PCI—a finding later corroborated in a larger cohort [142]. Systemic IL-6 levels begin to rise several hours after PCI and reach their peak at approximately 24 h [143,144]. In patients undergoing primary PCI, this peak is associated with the extent of myocardial necrosis [145]. IL-6 signalling has two routes: classical- or trans-signalling. Classical signalling involves binding of IL-6 to the IL-6 receptor expressed predominantly on hepatocytes. This pathway results in augmented production of acute-phase reactants such as CRP, fibrinogen and plasminogen [138]. Trans-signalling by IL-6 involves proteolytic cleavage of the IL-6 receptor, producing a soluble IL-6 receptor (sIL-6R) that can complex with IL-6 and bind to gp130, a transmembrane glycoprotein expressed almost ubiquitously throughout the human body [146]. This trans-signalling mechanism is responsible for the acute inflammatory response to PCI. In fact, a higher concentration of IL-6 and a lower concentration of sIL-6R in the coronary circulation following PCI is associated with an unfavourable prognosis [147]. This pattern of increased IL-6 in combination with reduced sIL-6R expression is indicative of trans-signalling, whereby a greater concentration of IL-6 depletes sIL-6R availability. Trans-signalling via sIL-6R is responsible for the recruitment and activation of both the innate and adaptive immune responses [138]. This rapid influx of leukocytes and release of pro-inflammatory cytokines within the culprit artery contributes to microembolisation, endothelial injury and platelet activation. Moreover, this augmented inflammatory response likely promotes plaque instability at non-culprit lesions and accelerates the progression of atherosclerotic disease—ultimately predisposing to future ischaemic events. 

The link between periprocedural inflammation and future heart failure is explained by a combination of factors. First, a greater degree of inflammation in the acute ischaemic period is associated with greater infarct size and thus reduced viable myocardium [148]. Second, IL-6 is a potent inducer of matrix metalloproteinase (MMP) expression, particularly MMP-9, which is known to contribute to cardiac fibrosis and remodelling [149,150]. Higher MMP-9 levels at the time of PCI have been associated with a greater risk of late-onset heart failure in acute MI patients treated with primary PCI [151]. Third, inflammation is a cardiodepressant. This concept was first highlighted in septic patients but is now well established in patients with decompensated heart failure. Elevated levels of IL-1β impair β-adrenergic signalling, myocardial energetics and cardiac contractility [152]. Conversely, inhibition of IL-1β signalling limits post-infarct heart failure by improving short-term cardiac function and reducing long-term remodelling and endothelial dysfunction [153].

The acute inflammatory response to PCI does not appear to be dramatically influenced by stent type. In an early study by Dibra et al. [154] participants were randomised to either a sirolimus-eluting or bare-metal stent and the change in CRP from pre- to post-PCI was calculated. The median change in CRP was 3.1 mg/L in the sirolimus group compared to 3.0 mg/L in the bare-metal group, representing no significant difference in the acute response to these two stents [154]. Similarly, Sardella et al. [155] measured IL-1β and IL-6 levels in the coronary sinus before and 20 min after implantation of either a bare, paclitaxel- or sirolimus-eluting stent. This study showed an acute rise in intracoronary concentrations of both cytokines, which was independent of stent type. Finally, by comparing the procedural rise in intracellular adhesion molecule-1, hsCRP and IL-6 between bare-metal and early generation drug-eluting stents, Sakr et al. [156] reported that the degree of PCI-induced inflammation is related to lesion characteristics as opposed to the type of stent. Although there is limited evidence on the acute inflammatory response to newer generation stents, such as bioabsorbable stents, we hypothesise the data will be analogous as the mechanism is driven by mechanical plaque disruption with minimal influence by stent properties. In saying this, stent properties are an important predictor of long-term inflammation and outcomes, though this topic is beyond the scope of the current review and has been extensively discussed elsewhere [157,158]. 

## 7. Treating Inflammation in the Periprocedural Period

Preprocedural statin and anti-platelet loading has been shown to effectively reduce the rate of periprocedural MI and future MACE—an effect which is mediated, at least in part, by the anti-inflammatory effect of these agents [116,159,160]. In fact, the anti-platelet therapy, clopidogrel, is most efficacious in patients with a high preprocedural CRP level [161]. Though the anti-inflammatory effect of different anti-platelet medications is not equal. A recent comparison of preprocedural clopidogrel vs. prasugrel vs. ticagrelor in ACS patients found that prasugrel was associated with the greatest increase in endothelial function and reduction in IL-6 concentration [162]—likely contributing to its superior efficacy for reducing postprocedural ischaemic events [159]. Given the efficacy of these routine medications, we hypothesise that the addition of a dedicated preprocedural anti-inflammatory agent will improve PCI outcomes, particularly for those with elevated inflammatory markers (i.e., ACS patients). Table 2 highlights those studies which have evaluated the efficacy of preprocedural anti-inflammatory treatment on PCI outcomes. 

### 7.1. Colchicine 

Colchicine is a potent anti-inflammatory medication currently used in the management of acute gout, familial Mediterranean fever and pericarditis [168]. The mechanism of action of colchicine is not completely understood, although it is known that colchicine binds to tubulin and inhibits microtubule polymerisation, a process essential for normal cellular function [169]. In the setting of atherosclerotic disease, colchicine has been shown to suppress activation of the NLRP3 inflammation in monocytes, inhibit monocyte migration and reduce neutrophil activation and NET formation [72,113,115,133,165]. 

Several, relatively small, clinical trials have evaluated the effect of preprocedural colchicine on short-term outcomes. Martínez et al. [93] in a cohort of 40 patients with ACS and 33 with stable CAD, demonstrated that 1.5 mg colchicine (1 mg followed by 0.5 mg 1 h later) administered up to 24 h prior to PCI reduced the transcoronary gradient of IL-1β, IL-18 and IL-6 immediately following PCI in ACS but not stable CAD patients. The transcoronary gradient is a validated tool to measure the local cardiac release of specific proteins and is calculated as the coronary sinus concentration minus the systemic arterial concentration of the protein of interest [170]. Using the same dosing and sampling regimen, our group has also reported that colchicine reduces the transcoronary concentration of MCP-1, chemokine ligand 5 and fractalkine [73], all of which have been previously implicated in the pathogenesis of atherosclerosis and atherothrombosis. Again, these findings were evident in ACS but not stable CAD patients. Most recently, we have demonstrated that the neutrophils of individuals with ACS are hyper-reactive to inflammatory stimuli and thus ‘primed’ to form NETs [113]. In the same study we showed that preprocedural colchicine administration (the same regimen as above) acts directly on these ‘primed’ neutrophils to inhibit intraprocedural NET formation by stabilising the neutrophil cytoskeleton. The anti-inflammatory effect of preprocedural colchicine is supported by results of the COLCHICINE-PCI trial [160]. The nested inflammatory biomarker sub-study of the COLCHICINE-PCI trial included 280 participants with both ACS and stable CAD. Colchicine administration was different to the aforementioned regimen with 1.2 mg given 1 to 2 h before coronary angiography, followed by 0.6 mg 1 h later or immediately pre-PCI (whichever came first). In this substudy, colchicine was shown to attenuate the median percent increase in IL-6 and hsCRP from baseline to 24 h post-PCI. Collectively, these studies clearly demonstrate that colchicine administration prior to PCI acutely suppresses local cardiac and systemic inflammation in the periprocedural period. 

Two studies have investigated the effect of colchicine on chronic inflammation following PCI. In the Low Dose Colchicine after Myocardial Infarction (LoDoCo-MI) study [171], which included 237 participants with an acute MI, 0.5 mg colchicine daily for 30 days was not associated with their primary endpoint of the proportion of patients with a residual CRP level < 2 mg/L at 30 days. However, there was a modest, albeit statistically non-significant, reduction in absolute CRP over this period. In another study of 44 participants with STEMI who underwent primary PCI [172], starting 1 mg colchicine daily shortly after PCI did not affect the peak CRP level during hospitalisation. The most likely reasons for the lack of positive results in these studies are two-fold. First, both studies were not adequately powered to detect small, yet possibly clinically significant reductions in CRP. Second, the dose and timing of colchicine administration may have been insufficient to elicit its full anti-inflammatory effect, with neither study including a loading dose of colchicine prior to PCI.

Despite its well-established anti-inflammatory effect, whether preprocedural colchicine reduces myonecrosis remains controversial. The first study to address this question was by Deftereos et al. [163] and included 151 patients with STEMI who underwent primary PCI. Colchicine was administered following diagnostic coronary angiography with a loading dose of 2 mg (1.5 mg followed by 0.5 mg 1 h later). Patients continued on 0.5 mg colchicine twice daily for 5 days. The primary outcome of the study was the area under the curve of CK-MB concentration over the 72 h after admission, which was reduced by almost 50% in the colchicine compared to the placebo group. Similarly, the median maximum high-sensitivity troponin value was >50% less in the colchicine group. In a subset of 60 participants, infarct size was measured by cardiac MRI at day 6–9 post-MI. In line with the biomarker results, colchicine treatment was associated with a marked reduction in absolute and relative infarct size. It is important to note that it is unclear whether this benefit is achieved by reducing MI-associated inflammation alone or contributions from both MI and PCI. The Colchicine in STEMI Patients Study (COVERT-MI) [173] is an ongoing prospective, randomised, double-blind trial designed to validate the earlier findings of Deftereos et al. with results expected Q3/Q4 2022. In contrast to Deftereos et al., the results of the full COLCHICINE-PCI study [160], which included a total of 400 participants (50% with ACS) found no effect of preprocedural colchicine loading on the incidence periprocedural myocardial injury, MI or 30-day MACE. Most recently, results of the colchicine to prevent periprocedural myocardial injury in percutaneous coronary intervention (COPE-PCI) pilot trial were published [82]. This placebo-controlled trial included 75 participants, of which 59% presented with NSTEMI and the remainder with stable angina. Although no participants developed periprocedural MI, when compared to placebo, colchicine (1 mg followed by 0.5 mg one hour later; 6–24 h pre-PCI) was shown to significantly reduce the incidence of periprocedural myocardial injury and the absolute change in high sensitivity troponin-I from pre- to post-PCI. When stratified by presentation type colchicine was shown to only be effective in those with NSTEMI, likely due to the higher inflammatory risk in this population [82]. The conflicting findings of the COLCHICINE-PCI and COPE-PCI trials may be explained by their differences in colchicine administration timing. In the COLCHICINE-PCI trial, colchicine was administered only 1–2 h prior to PCI, compared to the 6–24 h period in the COPE-PCI trial—suggesting that a longer lead-in time may facilitate the full effect of colchicine [82,160]. 

Finally, the potential benefit of treating postprocedural RIR with colchicine was highlighted in a secondary analysis of the COLCOT [174]. Briefly, this trial recruited 4661 participants with a recent MI, of whom 93% underwent PCI and were then randomised to either colchicine (0.5 mg daily) or placebo within 30 days of their index event [4]. In this secondary analysis, initiation of colchicine within 3 days of the index event was associated with the greatest reduction in the primary endpoint of MACE when compared to those who initiated colchicine >3 days after their index event [174]. Interestingly, when colchicine treatment was initiated >3 days after the index event there was no significant difference in the primary endpoint between colchicine and placebo, indicating the benefit of colchicine in the secondary prevention of MI is achieved largely by reducing inflammation in the acute postprocedural setting.

### 7.2. Tocilizumab 

Tocilizumab is a humanised monoclonal antibody that blocks the IL-6 receptor to reduce both classic and trans-signalling of IL-6. Tocilizumab is routinely used in a variety of rheumatological conditions [175]. To date there are only two small studies that have investigated the effect of tocilizumab on periprocedural outcomes. The first included 117 participants with NSTEMI and evaluated the impact of a single, pre-PCI, intravenous dose of tocilizumab on short-term inflammation and biochemical markers of infarct size [164]. HsCRP and high-sensitivity troponin were measured at seven consecutive time points over three days, allowing for calculation of the area under the curve. Tocilizumab was associated with a reduced area under the curve for both hsCRP and high-sensitivity troponin, indicating that IL-6 blockade prior to PCI limits myonecrosis in NSTEMI patients. To determine the therapeutic benefit of tocilizumab in the context of early reperfusion, the Assessing the Effect of Anti-IL-6 Treatment in Myocardial Infarction (ASSAIL-MI) trial was developed [176]. The ASSAIL-MI trial allocated 199 participants with STEMI to either a single, pre-PCI intravenous dose of tocilizumab or placebo and found that tocilizumab significantly increased their primary endpoint of myocardial salvage index, as measured by cardiac MRI at 3–7 days post PCI [165]. Furthermore, in the tocilizumab group the median final infarct size was 21% lower and the area under the curve of troponin-T during hospitalisation was 31% lower, although the study was underpowered to detect a statistically significant difference in these secondary endpoints [165].

Two additional studies have investigated the mechanism of tocilizumab in the aforementioned NSTEMI trial. The first study [166] measured coronary flow reserve and markers of endothelial activation in both groups. Unexpectedly, tocilizumab was found to not affect coronary microvascular or endothelial function, indicating an alternate anti-inflammatory mechanism is responsible for its beneficial effect. Though results of the ASSAIL-MI trial suggest otherwise [165]. More recently, proteomic analysis of participants in the same study yielded two novel findings [167]. Tocilizumab treatment was associated with reduced expression of the monocyte chemoattractant C-C motif chemokine ligand 23 (CCL23) and increased expression of proteinase 3, a serine protease expressed primarily by neutrophils. The increase in proteinase 3 is likely due to neutrophil apoptosis, resulting in neutropenia and in combination with reduced CCL23-induced monocyte chemotaxis would attenuate periprocedural inflammation. 

### 7.3. Anakinra 

Two small pilot studies evaluated the effect of anakinra, a recombinant IL-1 receptor antagonist, in the periprocedural period of patients with STEMI [177,178]. In both studies patients received a subcutaneous injection of anakinra (100 mg) daily for 14 days following STEMI with successful reperfusion. In a patient-level pooled analysis of these two studies with a median follow-up of 2 years [179], anakinra was found to have no effect on the recurrence of ischaemic events. However, anakinra was associated with a dramatic reduction in the incidence of new-onset heart failure, suggesting that IL-1 blockade immediately following reperfusion may improve infarct healing. These findings were supported by a recent phase 2 study, which demonstrated that anakinra (100 mg daily) commenced within 12 h of coronary angiography and continued for 14 days blunted the acute inflammatory response following STEMI and reduced the incidence of heart failure and heart failure hospitalisation over a median follow-up period of 12 months [180]. In a separate study of 182 participants with NSTEMI [181], of whom ~55% underwent PCI for their index event, anakinra was shown to dramatically reduce short-term inflammation. In this study subcutaneous anakinra (100 mg daily) was initiated shortly after presentation resulting in a reduced area under the curve for CRP within the seven days post-admission. Moreover, the absolute levels of CRP were lower in the anakinra group at both seven and 14 days. Despite this profound anti-inflammatory effect there was no difference in the area under the curve for troponin. As such, current evidence suggests that, although anakinra may acutely reduce inflammation in the periprocedural period, this effect does not translate to a reduction in myonecrosis. 

### 7.4. Novel Therapies

Selective removal of CRP from the circulation, known as CRP-apheresis, is a novel therapy for rapidly reducing inflammation. Despite being limited by its highly labour- and resource-intensive nature, early studies of CRP-apheresis in ACS show promising results. In a porcine model of acute MI, CRP-apheresis performed shortly after reperfusion reduced infarct size and improved left ventricular ejection fraction at 14 days post-MI [182]. CRP-apheresis has recently been trialled in a small, non-randomised study of 83 patients with STEMI [183]. In this study patients had two sessions of CRP-apheresis, with the first starting approximately 24 h after symptom onset and reducing plasma CRP concentration by roughly 50%. Those exposed to CRP-apheresis displayed marginal improvements in infarct size, ventricular function and microvascular obstruction, although these results were statistically indeterminant [183]. Further studies with sound methodology and robust clinical endpoints are required to formally evaluate the utility of CRP-apheresis. 

Beyond its role in the coagulation cascade, activated factor X (factor Xa) is thought to contribute to inflammation. In preclinical studies, factor Xa inhibitors, such as rivaroxaban, have been shown to attenuate atherosclerotic plaque progression and destabilisation by limiting inflammation [184,185,186]. Despite this, there are limited data on the efficacy of preprocedural factor Xa inhibitors in individuals without atrial fibrillation. This is likely due to the increased risk of major bleeding with these medications, which must be weighed against any potential anti-inflammatory benefit. 

## 8. Challenges to Treating Periprocedural Inflammation 

There are several challenges to designing and implementing an optimal anti-inflammatory regimen in the periprocedural period. First, identifying the most appropriate therapeutic target, with the goal being to minimise atherosclerosis-associated and PCI-induced inflammation without impairing immunity. In the CANTOS study [2], the beneficial anti-inflammatory effect of canakinumab was outweighed by its detrimental effect on immunity, resulting in a high rate of death from infections. Second, optimising the time of drug administration and dosage. To do this it must be established whether the inhibition of inflammation is of most importance prior to, during and/or after PCI. Third, identifying the population most likely to benefit from anti-inflammatory treatment. Intuitively this would be those patients with a high degree of baseline inflammation, such as those with ACS. Fourth, pinpointing the most appropriate endpoint for clinical studies. With the controversy surrounding the definition of periprocedural myocardial injury and MI and their prognostic significance, the use of more robust, hard clinical endpoints, such as 1-year mortality or MACE may be superior. Fifth, whether to continue anti-inflammatory treatment post-PCI. 

Future preclinical studies should aim to further dissect the mechanism(s) through which PCI elicits an acute inflammatory response. Results of such studies will facilitate a more targeted approach to anti-inflammatory treatment and guide future drug development. With the abundance of preclinical and observational data highlighting the maladaptive effect of periprocedural inflammation, further large-scale randomised trials, powered for hard clinical endpoints should be designed to formally evaluate whether preprocedural anti-inflammatory treatment improves short- and long-term outcomes in both stable CAD and ACS patients. 

## Figures and Tables

**Figure 1 cells-10-01391-f001:**
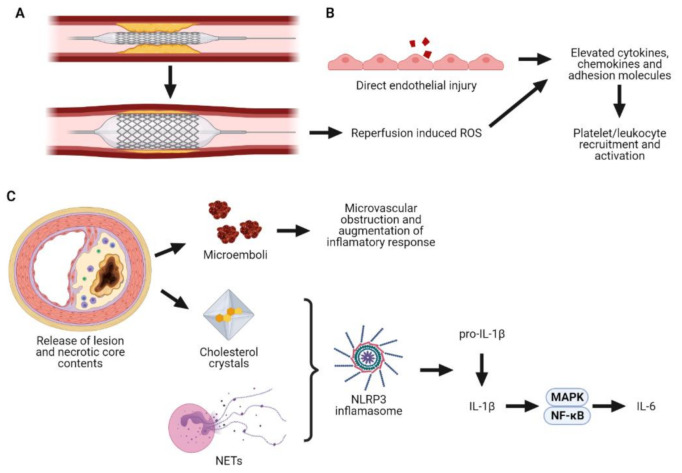
The inflammatory response to PCI. Balloon inflation and/or stent deployment (**A**) causes direct endothelial injury. In response the endothelium upregulates its expression of proinflammatory cytokines, chemokines and adhesion molecules, which drive platelet and leukocyte recruitment and activation (**B**). This process is further enhanced by reperfusion-induced ROS. Balloon inflation and/or stent deployment also causes microembolisation of plaque material, which travels downstream and contributes to microvascular obstruction. Moreover, the liberation of cholesterol crystals and NETs facilitates activation of the NLRP3 inflammasome, ultimately resulting in the production of IL-1β and IL-6 (**C**). IL-1β, Interleukin-1β; IL-6, Interleukin-6; MAPK, Mitogen-activated protein kinase; NETs, Neutrophil extracellular traps; NF-κB, Nuclear factor-κB; NLRP3, NOD-like receptor protein 3.

**Table 1 cells-10-01391-t001:** Association of preprocedural inflammation with clinical outcomes.

Population	Findings	Endpoint	Reference
CRP			
*n* = 500 SA, UA and NSTEMI	Preprocedural CRP > 3 mg/L associated with 2.4-fold higher incidence of PMI	PMI: CK-MB or troponin I > 3× URL	[28]
*n* = 1337 UA and NSTEMI	↑ Preprocedural hsCRP associated with ↑ risk of PMI and death	PMI: CK-MB > 5× URL	[29]
*n* = 96 SA and UA undergoing percutaneous bifurcation intervention	Positive linear relationship of preprocedural CRP with post procedural CK-MB	PMI: CK-MB > 3× URL	[30]
*n* = 85 SA	Preprocedural CRP > 6 mg/L associated with 2.5-fold higher incidence of PMI	PMI: Troponin I > 2 ng/mL	[31]
*n* = 4426 SA and UA	Preprocedural CRP > 3 mg/L associated with increased risk of PMI regardless of definition used	PMI: 2007 and 2012 universal definitions and SCAI definition	[32]
*n* = 463 SA, NSTEMI and STEMI	Positive linear relationship of preprocedural CRP with CK-MB ↑ CRP associated with 3-year MACE	PMI: CK-MB ≥ 3× URL	[33]
Retrospective: *n* = 7413 Prospective: *n* = 1189 SA and UA	Preprocedural CRP ≥ 3 mg/L, leukocyte count ≥ 7.3 × 10^9^/L and NLR ≥ 2.2 associated with increased risk of PMI	PMI: Troponin I > 5× URL	[34]
*n* = 1140 STEMI	↑ Preprocedural CRP independently associated with ↑ risk of no-reflow	Coronary no-reflow	[18]
*n* = 1217 STEMI	Preprocedural CRP, CRP to albumin ratio, leukocyte count and NLR were all independent predictors of no-reflow	Coronary no-reflow	[35]
*n* = 552 All with CTO	↑ Preprocedural hsCRP independently associated with ↑ risk of slow- and no-reflow	Coronary no-reflow	[36]
*n* = 167 SA on haemodialysis	↑ Preprocedural CRP associated with higher rates of MACE at 4 years and restenosis at 8 months	MACE and restenosis	[37]
*n* = 936 SA and UA	↑ Preprocedural CRP associated with higher incidence of primary endpoint at 2 years	Composite of death and Q-wave MI	[38]
*n* = 1650 SA and UA	↑ Preprocedural CRP associated with higher incidence of the primary outcome at 1 year	Composite of cardiac death and Q-wave MI	[39]
**Haematological parameters**			
*n* = 880 SA	↑ Preprocedural leukocyte count associated with ↑ incidence of PMI	PMI: CK-MB > 3× URL	[40]
*n* = 43 Patients undergoing carotid stenting	Preprocedural leukocyte count positively correlated with degree of intra-procedural microembolisation	Extent of distal embolisation	[41]
*n* = 99 STEMI	↑ Preprocedural leukocyte count associated with greater risk of no-reflow	Coronary no-reflow	[42]
*n* = 4450 SA and UA	↑ Preprocedural leukocyte count associated with greater 4-year mortality	All-cause mortality	[43]
*n* = 83 SA, UA and NSTEMI	Preprocedural leukocyte count and CRP were independent predictors of death and MI at 9 months	Composite of death and non-fatal MI	[44]
*n* = 909 SA, UA and NSTEMI	↑ Preprocedural eosinophil count favourable in short-term, yet detrimental in long-term	All-cause mortality	[45]
*n* = 1543 SA, UA and NSTEMI	No relationship of eosinophil count with occurrence of PMI	PMI: CK-MB ≥ 3× URL or an increase ≥ 50% if already elevated	[46]
*n* = 418 STEMI	Inverse relationship of TIMI flow grade with N/L ratio. ↑ Preprocedural N/L ratio associated with ↑ in-hospital MACE.	Coronary no-reflow and in-hospital MACE	[47]
*n* = 361 STEMI	↑ Preprocedural neutrophil count associated with higher incidence of no-reflow	Coronary no-reflow	[48]
*n* = 208 STEMI	TIMI frame count positively correlated with preprocedural neutrophil and platelet count, yet negatively correlated with lymphocyte count	Coronary no-reflow	[49]
*n* = 204 STEMI	↑ Preprocedural N/L ratio associated with no ST-resolution and greater 3-year mortality	Coronary no-reflow and 3-year all-cause mortality	[50]
*n* = 426 NSTEMI	↑ Preprocedural M/HDL ratio and ↓ L/M ratio associated with higher incidence of slow flow/no-reflow	Coronary no-reflow	[51]
*n* = 857 STEMI	↓ Preprocedural L/M ratio associated with higher incidence of no-reflow	Coronary no-reflow	[52]
*n* = 306 STEMI	↓ Preprocedural L/M ratio associated with greater short- and long-term MACCE	In-hospital and long-term MACCE	[53]
*n* = 83 STEMI	↑ Preprocedural platelet–neutrophil and platelet–monocyte aggregates associated with higher incidence of no-reflow	Coronary no-reflow	[54]
**Others**			
*n* = 50 STEMI	Preprocedural MPO concentration at culprit lesion directly correlated with TIMI frame count	Coronary no-reflow	[55]
*n* = 40 STEMI	↑ Preprocedural coronary MPO associated with greater post procedure microvascular obstruction	Microvascular obstruction	[56]
*n* = 192 STEMI	↑ Preprocedural MCP-1 associated with greater risk of no-reflow and 3-year mortality	Coronary no-reflow and all-cause mortality	[57]
*n* = 265 SA	Positive linear relationship between preprocedural Lp-PLA2 and postprocedural troponin T	PMI: troponin T > 20% baseline value and within 5× baseline value	[58]

↑: Increased; ↓: Decreased; CK-MB: Creatine kinase myocardial band; CMRI: Cardiac magnetic resonance imaging; CRP: C-reactive protein; Lp-PLA2: Lipoprotein-associated phospholipase A_2_; L/M: Lymphocyte to monocyte; MACE: Major adverse cardiac events; MACCE: Major adverse cardiac and cerebrovascular events; MCP-1: Monocyte chemoattractant protein-1; MI: Myocardial infarction; MPO: Myeloperoxidase; M/HDL: Monocyte to high-density lipoprotein cholesterol; NSTEMI: Non-ST-elevation myocardial infarction; N/L: Neutrophil to lymphocyte; PMI: Periprocedural myocardial infarction; SA: Stable angina; STEMI: ST-elevation myocardial infarction; TIMI: Thrombolysis in myocardial infarction; TLR: Target lesion revascularisation; UA: Unstable angina; URL: Upper reference limit.

**Table 2 cells-10-01391-t002:** Trials of preprocedural anti-inflammatory medications.

Population	Dose	Effect of anti-inflammatory	Reference
Colchicine			
*n* = 73 SA and ACS	1.5 mg pre-PCI	↓ Transcoronary concentration of IL-1β, IL-18 and IL-6 in ACS but not SA patients	[93]
*n* = 38 SA and ACS	1.5 mg pre-PCI	↓ Transcoronary concentration of MCP-1, CCL5 and fractalkine in ACS but not SA patients	[73]
*n* = 60 SA and ACS	1.5 mg pre-PCI	↓ Intra-procedural NET release within the coronary circulation	[113]
*n* = 400 SA and ACS	1.8 mg pre-PCI	No effect on incidence of PMI No effect on incidence of composite endpoint of death, non-fatal MI and target vessel revascularisation Suppressed post-procedural CRP and IL-6 elevation	[160]
*n* = 75 SA and NSTEMI	1.5 mg pre-PCI	↓ Periprocedural myocardial injury in NSTEMI patients	[82]
*n* = 151 STEMI	2 mg pre-PCI plus 0.5 mg twice daily for 5 days	↓ Area under the CK-MB curve during admission ↓ Infarct size	[163]
**Tocilizumab**			
*n* = 117 NSTEMI	280 mg prior to coronary angiography	↓ Area under the hsCRP curve during admission ↓ Area under the hsTnT curve during admission	[164]
*n* = 199 STEMI	280 mg prior to coronary angiography	↑ Myocardial salvage index at 3–7 post-PCI ↓ Microvascular obstruction	[165]
*n* = 42 NSTEMI	280 mg prior to coronary angiography	No effect on coronary flow reserve	[166]
*n* = 48 NSTEMI	280 mg prior to coronary angiography	↓ Lipopolysaccharide-binding protein, hepcidin, IGF-binding protein 4 and CCL23 ↑ Proteinase 3	[167]

↑: Increased; ↓: Decreased; ACS, acute coronary syndrome; CCL5, chemokine ligand 5; CCL23, chemokine ligand 23; CK-MB, creatinine kinase myocardial band; CRP, C-reactive protein; hsCRP, high sensitivity C-reactive protein; hsTnT, high sensitivity troponin T; IGF, insulin-like growth factor; IL-6, interleukin-6; IL-18, interleukin-18; Il-1β, interleukin-1β; MCP-1, monocyte chemoattractant protein-1; NET, neutrophil extracellular trap; NSTEMI, non-ST-elevation myocardial infarction; PMI, periprocedural myocardial infarction; SA, stable angina; STEMI, ST-elevation myocardial infarction.

## Data Availability

Not applicable.

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
