# Peer review of "Inflammation during Percutaneous Coronary Intervention—Prognostic Value, Mechanisms and Therapeutic Targets"

_cells, 2021, doi:10.3390/cells10061391_

Round 1

Reviewer 1 Report

Very interesting review paper. I have no major comments.

Author Response

We thank the reviewer for taking the time to read our manuscript and their kind comments. 

Reviewer 2 Report

Tucker et al. provide an overview and discussion on the occurrence, mechanisms, impact, and targeting inflammation related to percutaneous coronary intervention (PCI) in patients with coronary artery disease (CAD). This manuscript addresses an important and interesting topic, especially that inflammation associated with cardiovascular disease is currently a subject of extensive experimental and clinical investigation. Both the rationale and scientific content of the manuscript are valuable. However, I have several concerns that, in my opinion, deserve special attention in the revision.

I am not sure that the title of this review and the abstract are adequate given the objectives described in section 1 and the content of the manuscript. The title suggests that the review is focused on mechanisms of inflammatory response to PCI as a procedure in patients with CAD and potential therapeutic approaches targeting PCI-related inflammation for (implicitly) improving clinical outcomes. However, a large portion of the manuscript is devoted to occurrence and prognostic significance of periprocedural myocardial infarction (MI) per se (section 2), not necessarily in the context of inflammation. In addition, the extensive section 3 describes the aspects associated with a significance and prognostic value of preexisting preprocedural inflammation for cardiovascular risks rather than associations between performing PCI, inflammatory response, and subsequent consequences. Moreover, only one short section (section 5) is focused on clinical significance of postprocedural inflammation which can be both PCI-related and related to specific type of CAD in specific patients undergoing PCI. Also, the abstract does not represent adequately the content of the manuscript. For example, the first half of abstract is devoted to the epidemiology and pathophysiology of periprocedural MI rather than PCI-related inflammatory activation. These issues require attention.

Given the title and objectives of the manuscript included in section 1, one could expect that this review would include comprehensive data on mechanisms and implications of inflammation (intuitively both local and systemic) associated with PCI and anti-inflammatory strategies for reducing inflammation-related adverse clinical outcome. While the authors described in section 3 prognostic value of preprocedural inflammatory markers for atherosclerotic outcomes and death in short-term (with main focus on periprocedural MI) and long-term follow-up post-PCI in CAD patients, the topic on PCI-related postprocedural changes in inflammatory markers and their prognostic significance for cardiovascular outcomes requires more attention than is included in section 5. I suggest extending section 5 or adding another section to summarize these issues. In any event, the section 5 requires further attention and improvements. Also, I suggest to explicitly indicate which results regarding preprocedural and postprocedural changes in inflammatory markers refer to patients with stable CAD or patients with acute MI treated with PCI. This is important as post-PCI changes in inflammatory markers indicating an increased risk of adverse clinical outcomes, especially in patients with STEMI, can result from other mechanisms triggered by necrosis-related inflammation, and not only PCI-related inflammatory response.

In addition, while the authors included some results on the prognostic significance of inflammation (especially from preprocedural period) in patients with CAD treated with PCI for atherosclerotic adverse cardiac events and death, the data on the occurrence, mechanisms and usefulness of changes in inflammatory biomarkers post-PCI for the prediction of left ventricular (LV) dysfunction and remodeling, as well heart failure (HF) in CAD patients treated with PCI are omitted in this manuscript. The authors did not include information indicating that: (1) there is a clinical evidence that overactive and/or prolonged inflammatory activation (including ongoing post-infarct chronic low-grade inflammation) in the course of MI treated with PCI can contribute significantly to cardiac damage and dysfunction, and adverse clinical outcome including the development of HF, and (2) inflammatory biomarkers such as IL-6 and CRP can aid in identifying high-risk patients. Such information supports the need to continue basic and clinical research in this field. For example, it is of interest to establish if PCI that can trigger systemic inflammation contributes to further myocardial damage, endothelial dysfunction, and development of HF in patients with MI undergoing PCI despite other guideline-based therapies. Omitting these aspects in the manuscript represents an inconsistent approach because the authors included the results of a few clinical trials in section 6 which suggest that an inhibition of excessive inflammatory activation in MI is effective in a decrease in cardiac dysfunction and HF (L. 516-520). I suggest adding at least a paragraph in section 5 or a separate section devoted to this topic and including relevant references. Example relevant references that would fit in this context are:

Świątkiewicz, I., et al. C-Reactive Protein as a Risk Marker for Post-Infarct Heart Failure over a Multi-Year Period. Int. J. Mol. Sci. 2021, 22, 3169.

Abbate, A.; Toldo, S.; Marchetti, C.; Kron, J.; Van Tassell, B.W.; Dinarello, C.A. Interleukin-1 and the Inflammasome as Therapeutic Targets in Cardiovascular Disease. Circ. Res. 2020, 126, 1260–1280.

Swiatkiewicz, I. et al. Enhanced inflammation is a marker for risk of post-infarct ventricular dysfunction and heart failure. Int. J. Mol. Sci. 2020, 21, 807.

Everett, B.M.; Cornel, J.H.; Lainscak, M.; Anker, S.D.; Abbate, A.; Thuren, T.; Libby, P.; Glynn, R.J.; Ridker, P.M. Anti-Inflammatory Therapy With Canakinumab for the Prevention of Hospitalization for Heart Failure. Circulation 2019, 139, 1289–1299.

Toldo, S.; Abbate, A. The NLRP3 inflammasome in acute myocardial infarction. Nat. Rev. Cardiol. 2018, 15, 203–214.

Fanola, C.L., et al. Interleukin-6 and the Risk of Adverse Outcomes in Patients After an Acute Coronary Syndrome: Observations From the SOLID-TIMI 52 (Stabilization of Plaque Using Darapladib-Thrombolysis in Myocardial Infarction 52) Trial. J. Am. Heart Assoc. 2017, 6, e005637.

Świątkiewicz I., et al. I., Value of C-reactive protein in predicting left ventricular remodelling in patients with a first ST-segment elevation myocardial infarction, Med. Inflamm. 2012; 2012: 250867; doi:10.1155/2012/250867.

In section 4, the authors summarized pretty well the existing results of research and experimental studies on the role of various pathways involved in myocardial injury and inflammation in CAD including mechanisms of periprocedural MI post-PCI. However, the implications associated with the use of various types of stents including the newest generations require more attention. In addition, the mechanisms involved in inflammatory activation during the periprocedural period are different in patients with stable CAD and patients with acute MI, so I suggest describing these mechanisms for these two groups of patients separately. It would be also desirable to add a paragraph in section 4 about inflammation-related mechanisms triggered by MI treated with PCI which can lead to further myocardial injury, LV dysfunction, and HF. Although reperfusion limits the myocardial necrosis, release of intracellular cytokines and activation of NLRP3 contribute to local and systemic inflammatory responses that can enhance infarct size and promote HF.

Section 6 includes good description of data on the current status and future perspectives of anti-inflammatory strategies which have been proven in clinical trials. However, it is important to further improve section 6.4 to specify future directions of basic and clinical research on the topics related to inflammation and PCI in CAD patients and broader clinical implications of this kind of research because this topic is not clear enough in the present manuscript.

Reviewer 3 Report

In this review Tucker et all. Give an overview about periprocedural myocardial infarction and inflammation. Pathophysiology, clinical relevance and several targets are described. This topic is important as periprocedural inflammation contributes to early re-stenosis. However, some points need to be addressed.

  1. Please name the references at the end of the respective citation. It often remains unclear which reference you refer to, especially when there are two or more sources after one statement and none after the following. Examples are shown below.
    1. On page 2, line 55-65, reference needs to be added after the passage.
    2. On page 2, line 79-81 reference is missing.
    3. On page 4, line 131-132 please add how the association is (increased, decreased.).
    4. On page 4, line 145-148 reference is needed.
    5. On page 12, line 385 please introduce Canakinumab. Otherwise the link to previously described IL-1β remains unclear for readers.
    6. Page 13, line 411-416.
    7. Add reference on page 14, line 455.
  2. Page 15, line 481. Greatest reduction compared to what?
  3. On page 17, line 541 it is mentioned, that canakinumab’s positive effects were outweighed by increasing infections. Please further discuss this when you discuss CANTOS in “Postprocedural residual inflammatory risk”.
  4. Please consider reperfusion damage on inflammation in your discussion.
  5. Add p-values of the studies you mention in you tables when possible.
  6. Please discuss platelets and their role on cardiac inflammation. Further, please add, whether different platelet inhibitors have different effects on Inflammation.
  7. Please discuss, whether factor Xa inhibitors such as Rivaroxaban have a positive inflammatory effect.
  8. Please add CRP-apheresis to discussion.

Round 2

Reviewer 2 Report

The authors adequately addressed my comments.